# Isca, v1.0: A Framework for the Global Modelling of the Atmospheres of Earth and Other Planets at Varying Levels of Complexity

Geoffrey K. Vallis[1], Greg Colyer[1], Ruth Geen[1], Edwin Gerber[2],
Martin Jucker[3], Penelope Maher[1], Alexander Paterson[1],
Marianne Pietschnig[1], James Penn[1], and Stephen I. Thomson[1]

[1]University of Exeter
[2]New York University
[3]University of Melbourne

*Correspondence to:* Geoffrey K. Vallis (g.vallis@exeter.ac.uk)

**Abstract.**

   Isca is a framework for the idealized modelling of the global circulation of planetary atmospheres at varying levels of complexity and realism. The framework is an outgrowth of models from the Geophysical Fluid Dynamics Laboratory in Princeton, USA, designed for
Earth's atmosphere, but it may readily be extended into other planetary regimes. Various forcing and radiation options are available, from dry, time invariant, Newtonian thermal relaxation to moist dynamics with radiative transfer. Options are available in the dry thermal relaxation scheme to account for the effects of obliquity and eccentricity (and so seasonality), different atmospheric optical depths and a surface mixed layer. An idealized gray radiation
scheme, a two-band scheme and a multi-band scheme are also available, all with simple moist effects and astronomically-based solar forcing. At the complex end of the spectrum the framework provides a direct connection to comprehensive atmospheric general circulation models.

   For Earth modeling, options include an aqua-planet and configurable continental outlines
and topography. Continents may be defined by changing albedo, heat capacity and evaporative parameters, and/or by using a simple bucket hydrology model. Oceanic Q-fluxes may be added to reproduce specified sea-surface temperatures, with arbitrary continental distributions. Planetary atmospheres may be configured by changing planetary size and mass, solar forcing, atmospheric mass, radiative, and other parameters. Examples are given of various
Earth configurations as well as a giant planet simulation, a slowly-rotating terrestrial planet simulation, and tidally-locked and other orbitally-resonant exo-planet simulations.

The underlying model is written in Fortran and may largely be configured with Python scripts. Python scripts are also used to run the model on different architectures, to archive the output, and for diagnostics, graphics, and post-processing. All of these features are publicly available on a git-based repository.

## 1   Introduction

Understanding climate is not synonymous with predicting or simulating climate. In order to provide the best possible climate predictions of Earth's weather and climate we need comprehensive models that provide simulations with the greatest possible degree of verisimilitude. However, the development and use of such models does not necessarily lead to understanding nor, at a practical level, does it necessarily provide a path for the continued improvement of those models, as has been discussed extensively elsewhere (Schneider and Dickinson, 1974; Hoskins, 1983; Held, 2005; Vallis, 2016), and a hierarchical approach, and/or the use of models with different levels of complexity, is often advocated.

Consider also the atmospheres of other planets. The amount of data we have for the atmospheres of the planets of our own Solar System is orders of magnitude less than the data we have for Earth. And the amount of data we have for exoplanets is still orders of magnitude less than that. Yet over 4000 exoplanets are known to exist, and it is likely that there are, in fact, billions of such planets in our galaxy alone. To construct a comprehensive model for each of those planets would be foolish if it were not impossible. Rather, understanding will come through the use of more general principles governing the atmospheres, and possible oceans, of these planets, along with models that allow a much larger range of parameters than do comprehensive models of Earth's atmosphere. But much as we may laud the benefits of idealized models, they are of limited utility if they do not connect to the more comprehensive and realistic models that, we may hope, give us accurate simulations and connect to a real climate system or real planetary atmosphere. If there is no such connection then the idealized models may be solving the wrong problem and may simply be irrelevant. Evidently, there is no single level of complexity that is appropriate for all problems, and both simple and complicated models have their uses.

A variety of models at different levels of complexity have in fact been constructed. Thus, to name but a few, Fraedrich et al. (2005b); Frierson et al. (2006); O'Gorman and Schneider (2008); Blackburn and Hoskins (2013) and Joshi et al. (2015) all describe models of Earth's atmosphere that are simplified in some way compared to a full GCM (of which there are a very great many). Similarly, regarding planetary atmospheres and again giving a limited sample,

the Planet Simulator is a sibling of the PUMA model for planetary atmospheres (Fraedrich et al., 2005a); the SPARC model (Showman et al., 2009) uses the dynamical core of the MIT GCM but adds a more general radiation scheme appropriate for planetary atmospheres; the GFDL system has itself been used in a number of Earth and planetary settings (e.g., Mitchell et al., 2011; Schneider and Liu, 2009, others); the UK Met Office Unified Model has been configured in various ways for both terrestrial exoplanets and hot Jupiters (Mayne et al., 2014; Boutle et al., 2017); the THOR model (Mendonça et al., 2016) solves the deep non-hydrostatic equations (as does the Unified Model) on an icosohedral grid and is designed to explore a range of planetary atmospheres; and CliMT (https://github.com/CliMT/climt) aims to provide a flexible Python based climate modelling toolkit. A number of quite comprehensive models, targeted at specific planets and similar in some ways to full GCMs of Earth, have also been developed.

These models all have a range of different parameterizations and cover a wide range of circumstances, but it is hard to compare one to another and it is particularly hard to relate simple models to complicated models in a controlled fashion. It is the purpose of this paper to describe a framework, Isca,[1] that enables models of appropriate complexity to be constructed for the problem at hand in atmospheric circulation, or indeed the construction of a sequence of models of increasing complexity, with simpler models connecting seamlessly to more complex models in a true hierarchy. The first release of the Isca framework contains an atmospheric primitive equation model with a wide range of configurable options for thermal forcing and radiative transfer, continental and topographic configurations, and other atmospheric and planetary parameters. The framework uses the infrastructure provided by Flexible Modeling System (FMS, https://www.gfdl.noaa.gov/fms/) of the Geophysical Fluid Dynamics Laboratory (GFDL) in Princeton, USA, and in particular includes the models of Held and Suarez (1994), Frierson et al. (2006) and the MiMA model of Jucker and Gerber (2017). However, Isca both provides more options (e.g., continents, surface processes, different radiation schemes), as well as a straightforward means to configure those options and to set up and run experiments. A brief summary is provided below, with more detail given in subsequent sections. Many other options could be readily configured by the user.

1. A dry model with Newtonian thermal relaxation with:

   (a) A Held–Suarez thermal forcing (Held and Suarez, 1994).

---

[1]Isca is the name of a Roman city located where present-day Exeter (UK) is now. It is also the Latinized version of the Celtic word for running water. It seems that 'whisky' has the same root, namely 'Uisce'.

(b) A generalized thermal relaxation field, similar in latitudinal and height structure as the original Held-Suarez model, but with longitudinal variation producing differential day-side and night-side heating. The point of strongest heating is determined from the orbital and rotation rates of the planet, allowing for a custom diurnal cycle. The speed and direction of the forcing can be prescribed, including reverse direction (the sun rises in the west, sets in the east) and a tidally-locked configuration with a permanent day-side.

(c) A thermal relaxation field that is constructed from astronomical solar input and an approximate analytic solution to radiative-convective equations with a specified optical depth, lapse rate, radiative relaxation time and surface mixed layer depth. This allows the strength and extent of the seasonal cycle and height of the tropopause to be varied, still using relatively simple thermal forcing.

2. A moist model, with evaporation from the surface and fast condensation (that is, immediate precipitation and no explicit liquid water content in the atmosphere), interacting with radiation and convection as described below.

3. Various radiation schemes, including a gray scheme, as in Frierson et al. (2006); a gray scheme with moisture feedback, similar to Byrne and O'Gorman (2013); a two-plus-one-band (two infra-red, one solar) scheme with an infra-red window, similar to Geen et al. (2016); and a correlated-$k$ multi-band radiation scheme, the RRTM scheme described by Clough et al. (2005) and used in the MiMA model of Jucker and Gerber (2017). The radiation may be dependent on the model-predicted moisture levels or used with fixed optical depths in most of these schemes. The incoming solar radiation is calculated from astronomical parameters, and can vary from diurnally averaged to tidally-locked.

4. Various convective parameterizations, specifically a Betts–Miller convective relaxation (Betts, 1986; Betts and Miller, 1986; Frierson et al., 2007) and a simplified mass flux method, the relaxed Arakawa-Schubert or RAS scheme (Moorthi and Suarez, 1992). A simple dry scheme following Schneider and Walker (2006) is also available.

5. Continental land masses, using either a realistic continental outline (from ECMWF) or configurable idealized continents that are set up with Python scripts. The continents themselves may be defined by a changed heat capacity, albedo, surface roughness, evaporative parameters and/or a bucket hydrology model.

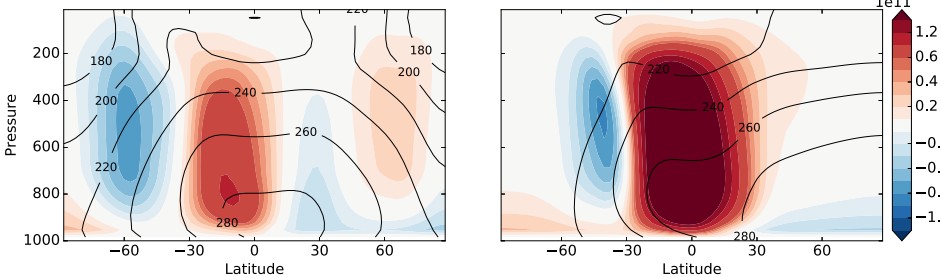

**Figure 1.** Meridional overturning circulation (colours, $10^{11}$ kg/s) and temperature (contours, K) in simulations with an obliquity of 10° (left) and 40° (right), at solstice, with Earth-like parameters otherwise, and a mixed layer depth of 10 m. (Earth's obliquity is 23.5°.) Note that at the higher obliquity the temperature is a maximum near the pole.

6. Horizontal heat fluxes — 'Q-fluxes' — that may be added to the ocean mixed layer to reproduce specified sea-surface temperatures. The algorithm may be applied with realistic continents, idealised continents or no continents.

7. Many parameters for other planetary atmospheres can be changed, including atmospheric mass, upper and lower pressure boundaries, planetary size and mass, planetary rotation rate, and choice of radiation scheme. All of the above can be done from a namelist or Python dictionary without recompilation.

8. The horizontal and vertical resolution of the model may be arbitrarily varied, although with a spectral core certain horizontal resolutions are preferable, for example T42, T63 or T213. Python software is available that enables a spin-up at low resolution and then an interpolation to and continued integration at higher resolution. A zonally-symmetric model – with no longitudinal variation but which can be used with most of the available 'physics' options – and a model that keeps only zonal wavenumbers 0, 1 and 2 are also configurable, and very fast compared to the full dynamical core.

In addition, we provide various Python scripts for configuring and running the model, archiving the output, producing various diagnostics and analyzing the results. The rest of the paper describes these options and how they may be implemented in more detail, and gives various examples. We provide a number of 'out-of-the-box' test cases, but in general it is up to the user to ensure that any model configuration is fit for purpose; with a framework such as this it is easy to configure a nonsensical planet. Our aim is not just to provide a ready-tuned intermediate model; rather, we provide a toolkit whereby the intelligent user may construct a

model or sequence of models, reasonably easily, for their own needs, be the models highly idealized or fairly comprehensive.

## 2 Model Foundations

The dynamical core of the framework is a spectral core from GFDL that uses sigma-pressure coordinates in the vertical. The code stems from that of Gordon and Stern (1982); it uses the spectral-transform methodology of Bourke (1974) and parallelizes using message passing without the need for shared memory. A very fast zonally-symmetric version of this dynamical core is available. It would be possible to use a grid-point dynamical core on a cubed sphere (from GFDL) but that configuration has not been implemented within Isca.

## 3 Options with a Dry Dynamical Core

In addition to the standard Held–Suarez benchmark (Held and Suarez, 1994) and its longitudinally-varying extension (item 2 above), we provide a more general thermal relaxation scheme that allows seasonal variation and possible extension to other planetary atmospheres. The essence of the scheme is as follows. We suppose that the atmosphere consists of a troposphere, with a given lapse rate, and a stratosphere that has a small optical depth and is in radiative equilibrium. Given also the optical depth of the atmosphere, then a radiative-convective tropopause height may be determined using the analytic formula of Vallis et al. (2015), namely

$$H_T = \frac{1}{16\Gamma}\left(CT_T + \sqrt{C^2 T_T^2 + 32\Gamma \tau_s H_a T_T}\right), \tag{1}$$

where $C = \log 4 \approx 1.4$, $\Gamma$ is the lapse rate, $T_T$ is the temperature at the tropopause, $\tau_s$ is the surface optical depth and $H_a$ is the scale height of the main infrared absorber. We determine $T_T$ at each latitude using an astronomical calculation based on the incoming solar radiation, which is a function of zenith angle, and so latitude, obliquity, time of year and solar constant. Note that this tropopause height will (correctly) increase if the optical depth increases, as with global warming, or if the specified lapse rate is made smaller.

Given the tropopause height, temperature and lapse rate, we then construct a radiative-convective relaxation temperature, $T_R$ as a function of height, latitude and time of year, using

$$T_R(y,z,t) = T_T(y,t) + \Gamma(H_T(y,t) - z). \tag{2}$$

This equation applied to the troposphere and may be extended upwards by assuming the stratospheric relaxation temperature is given by radiative equilibrium (other options also exist). We may then allow for the effects of a finite heat capacity of the surface by supposing that the ground temperature, $T_g$ obeys

$$C_g \frac{dT_g}{dt} = \sigma T_s^4 - \sigma T_g^4, \tag{3}$$

or a linearization thereof, where $C_g$ is the heat capacity of the surface (e.g., ocean mixed-layer or ground) and $T_s$ is the surface air temperature calculated using (2), integrating down from the tropopause to the surface with the specified lapse rate; that is, $T_s(y,t) = T_T(y,t) + \Gamma H_T$. We then use the calculated $T_g(y,t)$ from (3) and that same lapse rate to determine the radiative-

convective temperature at a height $z$, integrating up from the ground to the tropopause to give

$$T_R(y,z,t) = T_g(y,t) - \Gamma z. \tag{4}$$

This value of $T_R(y,z,t)$ is then used as the radiative-convective relaxation temperature instead of that given by (2), and is equal to it if $C_g = 0$. That is, the thermodynamic equation is

forced by a linear term $(T_R - T)/\tau$, where $\tau$ is a relaxation timescale (that might be chosen to be that given by Held and Suarez, or set by the user).

By virtue of having a finite surface heat capacity, the algorithm tempers the seasonal cycle and can ensure, for example, that the radiative-convective relaxation temperature is not absolute zero if the zenith angle is such that the incoming solar radiation is zero. Note

that the free-running model will determine its own tropopause height, through the combined effects of the thermal forcing and the model's own dynamics, and the resulting tropopause height may differ from that given by (1). (The differences will arise if there is meridional convergence of heat by the atmospheric dynamics or if the actual model lapse rate is different from $\Gamma$ in (1).)

By varying the obliquity, optical depth, surface heat capacity and atmospheric thermal relaxation time as needed we may obtain a wide range of seasonal cycles appropriate for Earth or other planets whilst keeping the simplicity of a dry dynamical core with a Newtonian thermal relaxation. A sample solution is shown in Fig. 1. This simulation uses Earth-like parameters — the rotation rate, equation of state, length of seasons and mass of

the atmosphere are all those of Earth (but all may be easily varied) — and with a mixed layer depth of 10 m. The panels both show the solsticial circulation and temperature, one with a 10° obliquity and the other with a 40° obliquity (Earth's obliquity is 23.5°). If the mixed layer depth were increased the seasonal cycle would be further tempered, and with

sufficiently high mixed layer depths both simulations converge to something similar to (but not exactly the same as) the Held–Suarez test case.

## 4   Radiation and Moist Model Options

The simplest moist model available uses gray radiation in the infra-red, a Betts–Miller type
convective relaxation scheme with no moisture feedback into the radiation, and a simple Monin–Obukhov boundary layer, as in the model of Frierson et al. (2006). The code for the boundary layer and convective schemes was provided by GFDL. Other radiative options are available as follows.

### 4.1   Moisture feedback with gray radiation

A simple scheme to incorporate moisture feedback is an extension of that introduced by Byrne and O'Gorman (2013). The scheme is gray in the infra-red so that a single optical thickness, $\tau$, is defined for the entirety of the longwave part spectrum, and includes a parameterization of longwave absorption by carbon dioxide, which we derived from Santa Barbara DISORT Atmospheric Radiative Transfer 60 (SBDART) output (Ricchiazzi et al., 1998). The optical
depth is calculated as a function of specific humidity, $q$ (kg/kg), the mixing ratio of carbon dioxide, $CO_2$ (ppm), and pressure, such that

$$\frac{d\tau}{d\sigma} = a\mu + bq + c\log(CO_2/360) \tag{5}$$

In the above, $\sigma = p/p_0$, i.e., pressure normalized by a constant ($10^5$ Pa), $a, b$ and $c$ are constants, and $\mu$, set to 1 as default, is a scaling parameter intended to represent absorption by
well-mixed gases. Byrne and O'Gorman (2013) used $a = 0.8678$ and $b = 1997.9$ and $c = 0$, with their coefficients based on fitting the above equation to the longwave optical depths of Frierson et al. (2006). For experiments with an albedo closer to that of Earth than was used in their idealised study ($\approx 0.3$ vs $\approx 0.38$), we suggest values of $a = 0.1627, b = 1997.9$, and $c = 0.17$. However, these are easily changed by the user. In the shortwave, the optical
depths of Frierson et al. (2006) may still be used, or all shortwave radiation may be assumed absorbed at the surface in the simplest case.

This scheme provides a simple tool for experiments in which only a lowest order description of water vapour radiative feedback is required. A limitation of the above gray scheme is that in reality the longwave absorption spectra of water vapour and carbon dioxide are far
from uniform, so that the scheme captures only the very basic structure of the longwave

radiative heating. The next step up in complexity is to use two bands in the infra-red, as we now describe.

## 4.2 Simple radiation with an infra-red window

To provide an intermediate option between gray radiation and a more complete description of radiative transfer, a scheme with two infra-red bands and one solar band, as described in Geen et al. (2016), has been incorporated into our model with some adjustments.[2] The shortwave band ($< 4\mu m$) treats all solar radiation and the two longwave bands treat absorption in the infra-red window region of the spectrum (8–14 $\mu m$), and in all other longwave wavelengths ($> 4\mu m$, non-window), respectively. All bands were originally parameterized by fitting to data from SBDART for a range of atmospheric profiles. Differences from Geen et al. (2016) are the addition of $CO_2$ absorption in each band, and changes to the functional form of the non-window optical depth formula. Although the original functional form was adequate with fixed sea surface temperatures, it was found to be unstable when coupled to a mixed layer ocean. An alternative form has therefore been fitted, which uses a log function rather than a power law to relate specific humidity to optical depth. The resultant parameterization is, for the shortwave,

$$\frac{d\tau^{sw}}{d\sigma} = a_{sw} + b_{sw}(\tau^{sw})q + c_{sw}\log(CO_2/360) \tag{6a}$$

where

$$\log(b_{sw}(\tau^{sw})) = \frac{0.01887}{\tau^{sw} + 0.009522} + \frac{1.603}{(\tau^{sw} + 0.5194)^2} \tag{6b}$$

and for the longwave,

$$\frac{d\tau^{lw}}{d\sigma} = a_{lw} + b_{lw}\log(c_{lw}q + 1) + d_{lw}\log\frac{CO_2}{360}, \tag{7a}$$

$$\frac{d\tau^{win}}{d\sigma} = a_{win} + b_{win}q + c_{win}q^2 + d_{win}\log\frac{CO_2}{360} \tag{7b}$$

Suggested values of the coefficients are given in the model documentation. Given these optical depths, two-stream equations are used to obtain the irradiances which are then weighted by

---

[2]Atmospheric radiation models nearly always treat solar radiation and infra-red radiation separately. In keeping with common usage, we will refer to models that have one solar band and one infra-red band as 'gray', as they are gray in the infra-red. Consistent with that, the scheme with two longwave bands and one solar band will be referred to as a 'two-band', or a 'two-plus-one band' scheme.

the Planck function for the bands in question. Thus, for the long-wave non-window region,

$$\frac{\mathrm{d}U^{lw}}{\mathrm{d}\tau^{lw}} = U^{lw} - B^{lw}, \qquad \frac{\mathrm{d}D^{lw}}{\mathrm{d}\tau^{lw}} = D^{lw} - B^{lw},$$
$$B = R^{lw}\sigma T^4,$$

(8)

and for the window,

$$\frac{\mathrm{d}U^{win}}{\mathrm{d}\tau^{win}} = U^{win} - B^{win}, \qquad \frac{\mathrm{d}D^{win}}{\mathrm{d}\tau^{win}} = D^{win} - B^{win},$$
$$B^{win} = R^{win}\sigma T^4,$$

(9)

where $R^{lw}$ and $R^{win}$ are the fractional irradiances in the non-window and window regions. These are configurable parameters with default values of 0.63 and 0.37.

The longwave heating rates calculated using this scheme give a notably improved accuracy for Earth's atmosphere over the gray schemes described in the previous section (Fig. 2), and although not as accurate as a full radiative transfer code the scheme is many times faster, enabling very long integrations to be carried out. Furthermore, the scheme is very configurable and tunable, and could allow for the simulation of other planetary atmospheres where the compositions are not accurately known (and so a complicated scheme is not warranted) and/or where a gray scheme fails (for example, a gray atmosphere is overly prone to a runaway greenhouse since radiation from the surface finds it too hard to escape without an infra-red window).

### 4.3 A full radiation scheme and the MiMA model

The most accurate radiative scheme in the current suite of options uses the multi-band correlated-$k$ Rapid Radiative Transfer Model (RRTM), described in Mlawer et al. (1997) and Clough et al. (2005). (The 'correlated-$k$' method, with $k$ being the absorption coefficient, is a means to efficiently calculate radiative transfer over a broad spectral range by collecting together wavenumber intervals with similar spectral properties and by supposing that these spectral properties are correlated from one level to another. A relatively small set of absorption coefficients can then be chosen to be representative of the absorption coefficients for all frequencies, leading to an enormous speed-up over line-by-line calculations and much better accuracy than traditional band methods that more simplistically just group together similar wavenumbers.) The implementation of this scheme largely follows that of Jucker and Gerber (2017) in the MiMA model, an aquaplanet model with simple topography. Within Isca the RRTM scheme may also be configured with idealized or realistic continental outlines and topography, a diurnal and seasonal cycle, or used with solar inputs appropriate for other

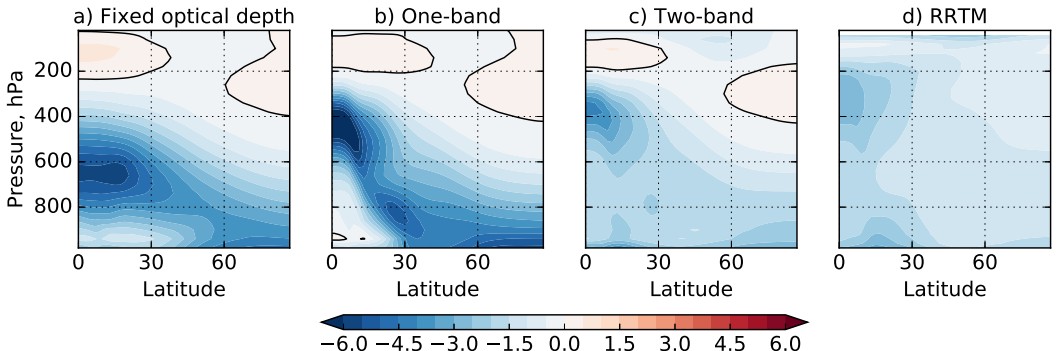

**Figure 2.** Longwave heating rates (K/day) for some of the radiation schemes available in Isca, for the given temperature and specific humidity fields shown in Fig. 3. The leftmost panel shows results with a gray scheme with a fixed optical depth, a function only of pressure and latitude, as in Frierson et al. (2006). The 'one band' scheme is also gray, but has an optical depth that is a function of water vapor and $CO_2$. The two-band scheme has two infra-red bands, and the RRTM scheme is a full, multi-band scheme, and both have and water vapour and $CO_2$ dependence.

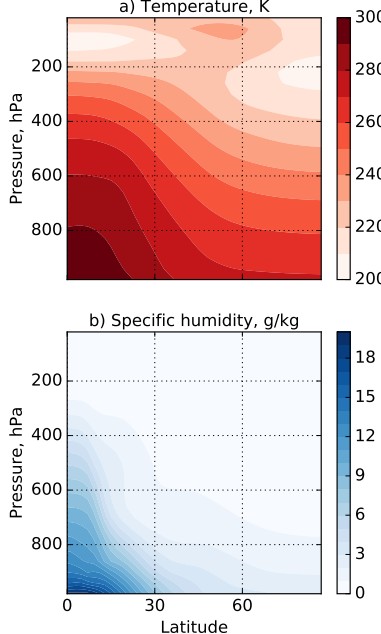

**Figure 3.** The input temperature and humidity profiles used in the radiation schemes shown in Fig. 2.

planets, as may all the radiation schemes in the framework. The RRTM scheme we use was primarily developed for Earth's atmosphere or variations about it, for which it is very accurate. It allows configurable levels of $CO_2$ and ozone, and it enables the model to produce a stratosphere and polar vortex. In principle the scheme could be re-calibrated to planetary atmospheres with different compositions and host stars with different emission spectra if the appropriate spectral files (*k*-distributions) were available.

The upper boundary of Isca may be specified by the user, and a user-configurable sponge layer and gravity-wave parameterization are available, so that with RRTM a true 'high-top' model is in principle available. However, in practice such things as the breaking of gravity waves at very high altitudes may lead to numerical difficulties and such a model may not perform satisfactorily out of the box, without some experimentation by the user.

### 4.4    Sample results with the various radiation schemes

Some sample results with the various radiation schemes are shown in Fig. 2, which shows the longwave cooling rate as a function of latitude and height for a given distribution of temperature and moisture, shown in Fig. 3. (All of these schemes may be used off-line, with a Python interface, although this is not currently part of the Isca repository.) The RRTM scheme gives very similar results to the SBDART scheme (not shown), and is the most accurate of our collection for Earth parameters. With the parameters chosen, the two-band scheme is more accurate than either of the two gray schemes, although it is possible that the gray schemes could be further tuned to match the RRTM results. However, we do not regard improved accuracy as the main advantage of the two-band scheme; rather, the presence of an infra-red window is a qualitative improvement over a gray scheme when more extreme climates, or other planetary atmospheres, are to be explored.

### 5    Aquaplanets and Continents

Isca has the ability to include continents that can either have a realistic geometry or a very idealized one (for example, a square continent) or something in between. Creating land-sea contrast within the Isca framework is a two-stage process. The first stage is the creation of a land-mask that defines the continent shapes and locations, and the second stage is the choice of how the properties of the surface should differ between land and ocean. In Isca, land is either essentially treated as a mixed-layer ocean but with various different heat capacity,

albedo and evaporative parameterizations, or we can include a simple bucket hydrology model described below.

## 5.1 Configuring continental outlines

Python software is provided to create a land-sea mask, which is an array of ones and zeros defining where land is, and where it is not, respectively. Such a mask is defined on the latitude-longitude grid of the model at the specified horizontal resolution. The Python software will output this array as a NetCDF file, which the model itself will take as an input file. Options within this software for different continent shapes include using realistic continental outlines taken from the ERA-interim invariant dataset (Dee et al., 2011), the simplified continental outlines similar to those of (Brayshaw et al., 2009; Saulière et al., 2012) with or without additions such as India and Australia, and simple rectangular continents defined using latitude and longitude ranges, all easily configurable by the user. Examples of integrations with idealized and realistic continental outlines are given in Fig. 4, Fig. 5 and Fig. 7.

## 5.2 Differentiating continents from ocean

Once a land-sea mask has been created, the Isca framework has options for using this mask to alter properties of the model's mixed-layer ocean. The properties that can be altered in regions of land are the depth of the mixed layer (i.e., the heat-capacity of the surface in regions of land), the surface albedo, the 'evaporative resistance' of the surface, and the roughness-length seen by the boundary-layer scheme. Evaporative resistance parameters ($\beta$ and $\alpha$) are used in the bulk formula for surface evaporation flux, $E$, so that

$$E = \rho_a C |v_a| \beta (\alpha q_s^* - q_a). \tag{10}$$

Here $\rho_a$ and $q_a$ are the atmospheric density and specific humidity in the lowest model layer, and $q_s^*$ is the saturation specific humidity calculated using the surface temperature (see e.g., equation (11) in Frierson et al. (2006)). The parameters $\beta$ and $\alpha$ are chosen by the user. Typically, one of them might be unity and the other lie between 0 and 1, and such values will reduce evaporation from a region of land, as would be evident in the real world. Using $\alpha = 1$ and $\beta < 1$ has the advantage of not allowing $E$ to change sign from what it would have been had $\alpha = 1$, and this formulation is normally chosen when using the bucket model, described below. We have tested both formulations in an Earth-like control case and found the differences to be small. When $\beta = \alpha = 1$ then the evaporation is equal to the 'potential evaporation', $E_0 = \rho_a C |v_a| (q_s^* - q_a)$.

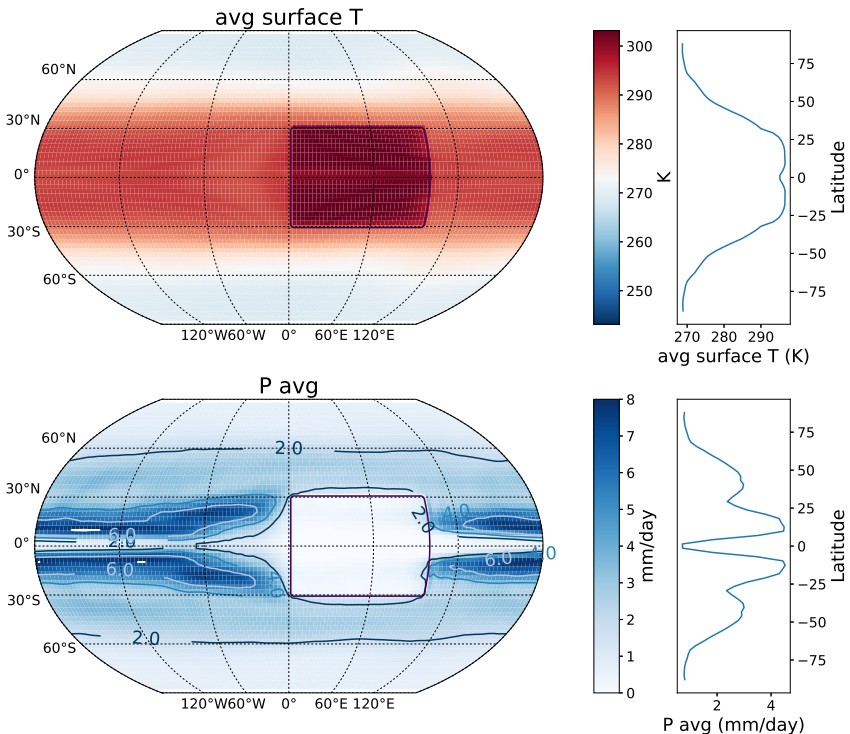

**Figure 4.** Annually-averaged temperature (top) and precipitation (bottom), with zonal averages shown in the right-hand panels. This model has an idealized, flat, rectangular continent, clearly visible, seasons, an obliquity of 23°, and uses Q-fluxes that target zonally-averaged AMIP sea-surface temperatures derived from Taylor et al. (2000). The ocean has a heat capacity of a 20 m mixed-layer depth and the land has a heat capacity equivalent to 2 m.

### 5.3 Topography

Since the dynamical core uses pressure-sigma coordinates implementing bottom topography is straightforward, as first described by Phillips (1957) and implemented by Gordon and Stern (1982) in a similar dynamical core. Within Isca the incorporation of topography simply involves specification of a topographic field $\eta(\lambda, \vartheta)$ — that is, height as a function of longitude and latitude. The topography may be either idealized – as, for example, implemented by Gerber and Vallis (2009) – or be taken from cartography in a NetCDF file. The topography used in the left-hand panel of Fig. 7 uses a realistic topography taken from the ECMWF interim data set (Dee et al., 2011), whereas Fig. 5 has no topography. In any case, topographic fields are easily constructed by the user and may be applied in other planetary configurations or even over the ocean. A Python script may be used to specify topography, just as in the

continental case, which writes out a netCDF file. Various topographic configurations are already available in this script, for example Gaussian mountains at specified locations, or topographies similar to those of Saulière et al. (2012), and others may be constructed by the user. A flag is available to set the topographic height to be zero over the ocean if desired –

without it, a Gaussian mountain over land would lead to non-zero topography over the ocean.

The user should be aware of potential inaccuracies in using steep topography in sigma co-ordinates (Haney, 1991), such as might be encountered on Mars (although mitigated there by the low gravity), and of potential Gibbs effects ('ringing') when using sharp topography in a spectral model (e.g., Navarra et al., 1994). For these reasons the topography may have to

be smoothed in some instances, for which functionality is provided in Isca's Fortran code.

## 5.4   A bucket hydrology

As an alternative to using a prescribed evaporative resistance to describe the differences in surface latent heat flux over land and ocean, a 'bucket model' similar to that of Manabe (1969) (also used in the idealized set ups of Farneti and Vallis 2009, and Liu and Schneider

2016) is included in Isca. Over land, soil hydrology is taken to be described by a bucket, which can be filled by precipitation, or emptied by evaporation. At any time the bucket depth, $W$, is between 0, corresponding to an empty bucket, and its field capacity, $W_{FC}$, corresponding to a full bucket. When the bucket is empty there can be no evaporation, and in general evaporation is proportional to the bucket depth as a fraction of the field capacity.

Bucket depth may not exceed field capacity so that when the bucket is full any net moisture flux into the bucket is treated as run-off, and does not increase the bucket depth. The default field capacity over land is set as 15 cm, but this is configurable.

The equations used to describe this behaviour over land are:

$$\frac{\mathrm{d}W}{\mathrm{d}t} = P - \beta E_0 \quad \text{if} \quad W < W_{FC} \quad \text{or} \quad P \le \beta E_0 \tag{11a}$$

$$\frac{\mathrm{d}W}{\mathrm{d}t} = 0 \qquad \text{if} \quad W = W_{FC} \quad \text{and} \quad P > \beta E_0, \tag{11b}$$

where $\beta$ is the parameter in (10), $P$ is precipitation, $E_0$ is the potential evaporation, given by (10) with $\beta = \alpha = 1$, and where, to give one example,

$$\beta = 1 \qquad \text{if} \qquad W \ge 0.75 W_{FC} \tag{11c}$$

$$\beta = \frac{W}{0.75 W_{FC}} \qquad \text{if} \qquad W < 0.75 W_{FC}. \tag{11d}$$

The parameters in these formulae are easily configurable and the oceans effectively have an infinite bucket depth, with $\beta = 1$ at all times. Some results using a bucket model in a

somewhat extreme case with a very idealized and rather large, rectangular, tropical continent are shown in Fig. 4.

## 6 Ocean Heat Fluxes

With a mixed-layer ocean having no dynamical heat transport, Earth-like climates are difficult
to obtain when a seasonal-cycle in insolation is included. This is because the position of the latitudinal maximum in surface temperature, as calculated in the model, lags behind the maximum of the insolation more than is observed in reality unless a very small mixed layer depth ($\sim$ 2 metres) is used. A lack of realism is also evident in simulations run with perpetual equinox insolation, with the lack of ocean heat transport forcing the atmosphere to
transport more heat poleward than it would in reality, particularly in the tropics where the Hadley Cell becomes too strong. Given these deficiencies, a so-called 'Q-flux' is added to the mixed-layer ocean temperature equation,

$$C_m \frac{\partial T}{\partial t} = \text{SW} + \text{LW} - \text{Sensible} - \text{Latent} + \nabla \cdot \boldsymbol{Q}. \tag{12}$$

Here $C_m$ is the mixed-layer's heat capacity, $T$ is surface ocean temperature, $t$ is time, 'SW'
and 'LW' are the net short-wave and long-wave radiative fluxes, respectively. 'Sensible' is the sensible heat-flux, 'Latent' is the latent heat flux, and $\boldsymbol{Q}$ is the Q-flux, being a two-dimensional vector that represents horizontal heat transport due to ocean dynamics. In equinoctial or annually-averaged cases an analytic formula for the Q-flux might be used to distribute heat in latitude, but such a formulation is difficult to adapt to problems with seasonally-varying
insolation. To overcome this problem, we have implemented a Q-flux method following Russell et al. (1985). This method uses several model integrations to calculate what the Q-flux needs to be in order to have the model's mixed-layer temperatures look like a set of specified input temperatures, as described below.

### 6.1 Calculation of Q-fluxes

1. An annually-repeating climatology of sea-surface temperatures must first be created. This could be from observations, or from AMIP SST data, or from some other source. Python software is provided for doing this.

2. Using the SST data as an input file, a chosen model configuration, with any continental configuration, is run with the prescribed SSTs (i.e. without the interactive SSTs of the

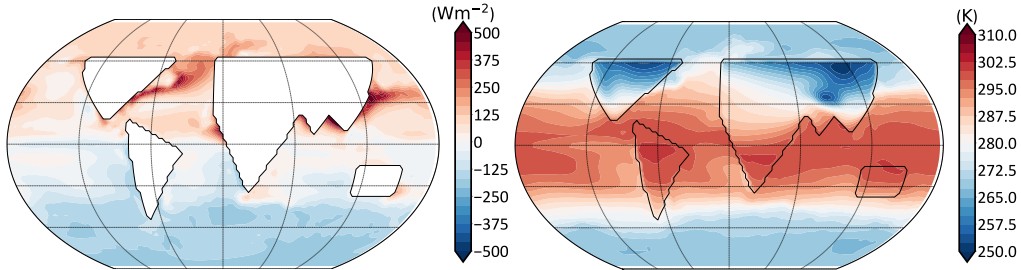

**Figure 5.** (a) The December-January-February (DJF) mean Q-flux divergence ($\nabla \cdot \mathbf{Q}$) calculated in a control case with a simple distribution of continents with a fixed evaporative resistance. (b) The resulting surface temperature, again in DJF, time-averaged over 20 years.

mixed-layer ocean, but still retaining its surface flux calculations). From this run, a climatology of surface fluxes can be calculated.

3. The climatology of surface fluxes, along with the input SST data itself, is used to calculate the Q-fluxes necessary to keep the free-running mixed-layer ocean's SSTs close to the SSTs prescribed in step 2. Python software is also provided for this calculation. The software outputs such Q-fluxes into a NetCDF file, which can then be used as model input. The integral of the Q-flux divergence is zero, so that the overall ocean temperature can respond to changed radiative conditions.

4. Having calculated these Q-fluxes, the model can be run using the mixed-layer ocean with the seasonally-varying Q-fluxes read from an input file. An example of the $\nabla \cdot \mathbf{Q}$ field calculated using this method is given in figure 5a, in the case with simplified continent outlines. The resulting SST field is shown in 5b.

This method was used within Isca by Thomson and Vallis (2018) and by Geen et al. (2018) to keep the model's mixed-layer temperatures close to a climatology of the sea-surface temperatures taken from the AMIP SST dataset (Taylor et al., 2000).

## 6.2 Ice

Isca also includes a very simple representation of sea and land ice, primarily designed for water ice on Earth. The representation is a passive representation, meaning the ice distribution is prescribed and does not depend on any changes in atmospheric or oceanic temperature. Regions of ice and non-ice are defined using an input dataset of ice-concentration (values between 0 and 1), which can be time-varying or constant-in-time. The model's representation

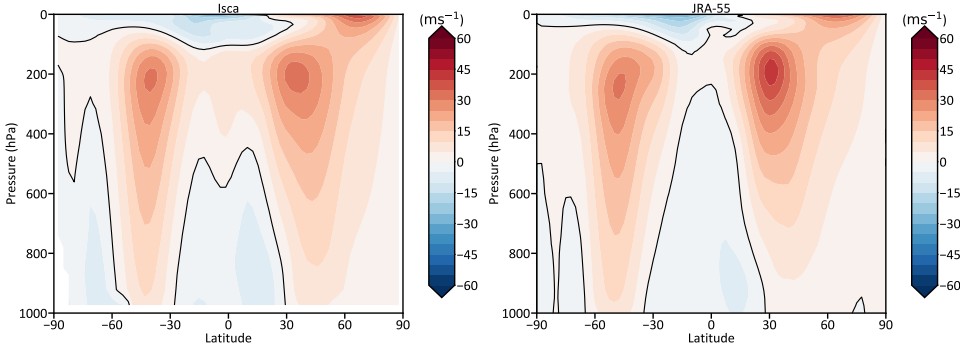

**Figure 6.** Zonal mean zonal wind in Isca (left) and from a re-analysis, JRA-55 (Kobayashi et al., 2015, right). The Isca results are an average over 20 years with parameters as described in the text, and JRA-55 shows an average between 1958 and 2016. The thick black line is the zero contour.

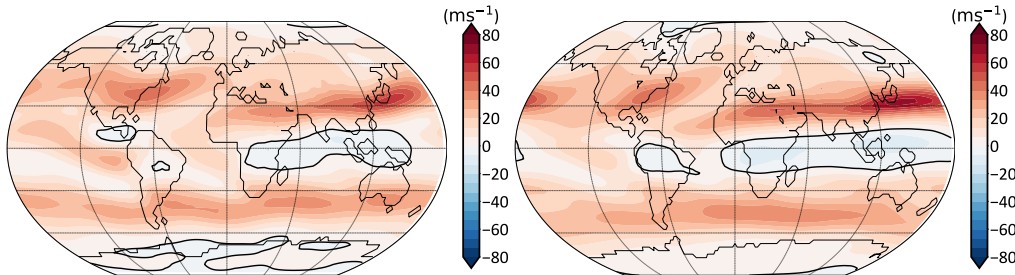

**Figure 7.** As for Fig. 6, but showing the zonal wind at 250 hPa, with Isca results on the left and the JRA-55 re-analysis on the right. The thick black line is the zero contour.

of ice is then binary, with a region either having ice, or no ice. The regions of ice are decided using an configurable ice-concentration threshold, with values above the threshold in the input dataset considered as ice, and those below the threshold considered as having no ice.

5     In regions of ice, the model's surface albedo is set to an ice-albedo value, which is also an input parameter. In regions of ice that are over ocean, the ocean Q-flux is set to zero with other properties of the surface remaining unchanged, with regions of land having the original land surface heat capacity and regions of ocean having the original ocean heat capacity.

    Including this representation of ice is particularly advantageous over the poles during the summer season, where the high ice albedo leads to much colder, and hence more realistic,

10  surface temperatures than if the standard land or ocean albedo is used in these regions (not shown).

## 7    Some Results

We now show various results of using Isca for Earth configured fairly realistically. Specifically, we use a full radiation scheme (RRTM) with $CO_2$ levels of 300 ppm and an ozone distribution taken from Jucker and Gerber (2017), a realistic distribution of continents and topography, seasonally varying ocean Q-fluxes that target an AMIP sea-surface temperature climatology (Taylor et al., 2000), and the simple ice model where regions with ice concentrations over 50% are given an albedo of 0.7. The ice concentration data was calculated as an annual mean, and mean over all years, of the AMIP ice input datasets of (Taylor et al., 2000). This configuration leads to the results shown in Fig. 6 and Fig. 7.

Of course, many comprehensive models, such as those submitted to the CMIP5 archive, can produce equally or more realistic results. Rather, our intent here is to show that the same model framework can pass in a near-continuous fashion from being highly idealized (as for example, in Fig. 1) to producing results similar to observations.

## 8    Planetary Atmospheres

Atmospheres of other planets may be configured by changing many of the parameters and configuration options described above. Here we give three examples of planetary configurations: a giant planet simulation with moisture and radiation; a slowly-rotating planet with a deep atmosphere simulated with a dry dynamical core; and two exoplanet cases, one tidally-locked and the other not.

### 8.1    Giant planets

Giant planet models may be configured with Isca, provided that the thickness of the modelled atmosphere is small compared to the planetary radius. For example, one relatively simple giant planet model, available as a pre-configured test case in Isca, draws from the Jupiter model described in Schneider and Liu (2009), from which it takes a gray radiation and dry convection scheme. The bottom boundary of this case (at 3 bars) has no mixed-layer surface but energy conservation is enforced whereby the upward thermal radiative flux is set equal to the sum of the downward solar and thermal fluxes at the surface. Also at the surface, a spatially-uniform heating is added in the bottom level of the atmosphere, which is used to represent heat emanating from the planet's interior. In the test case we turn off all sources and sinks of moisture, although adding moisture is a reasonably simple extension. Instead of a boundary-layer scheme, a Rayleigh drag is applied at the model's bottom boundary to

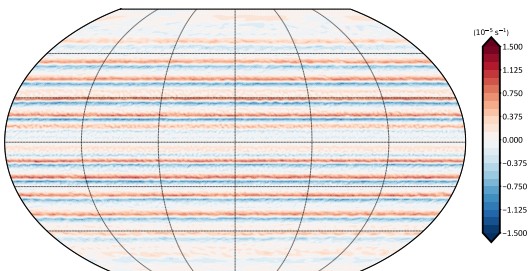

**Figure 8.** Time-averaged relative vorticity plotted on the 500 hPa surface, taken from a giant planet simulation with Isca, as described in the text. Multiple zonally-symmetric zonal jets are visible. Time-averaging is over 720 Earth days.

represent dissipative processes in the interior. This drag extends over all latitudes in the test case but can also be applied only over a chosen range of latitudes.

We also provide a drag formulation that can be applied at different levels within the atmosphere, rather than just at the model's bottom boundary. This is motivated by the results of Thomson and McIntyre (2016), who suggest that the effects of moist convection on Jupiter can be thought of as a Rayleigh drag near the water-cloud level ($\sim$ 1 bar in pressure), rather than the Rayleigh drag often used at the bottom boundary of many GCMs. The equation for this drag is

$$F_{\text{drag}}(\vartheta, \lambda, \sigma) = -r(\sigma)u(\vartheta, \lambda, \sigma), \tag{13}$$

where $\vartheta$ and $\lambda$ are latitude and longitude, respectively, $\sigma = p/p_{\text{surf}}$ is the standard terrain-following $\sigma$ coordinate, and $r$ is the drag coefficient. In our formulation, this coefficient takes the form

$$k(\sigma) = \begin{cases} \dfrac{1}{\tau_d}\max\left(0, \dfrac{\sigma - \sigma_t}{\sigma_m - \sigma_t}\right) & \sigma_t < \sigma < \sigma_m \\[2ex] \dfrac{1}{\tau_d}\max\left(0, \dfrac{\sigma_b - \sigma}{\sigma_b - \sigma_m}\right) & \sigma_m < \sigma < \sigma_b \end{cases} \tag{14}$$

with $\sigma_b$ is the lowest level at which the drag is applied at, $\sigma_t$ is the top level at which the drag is applied, and $\sigma_m$ is the level at which the drag is maximum. Using this drag formulation, and having the drag centered at 1 bar in pressure, the model produces overturning cells that only extend from the top of the model to the level of drag at 1 bar, rather than throughout the depth of the model. A 2D map of the vorticity at 0.5 bar, with drag centered at 1 bar, is shown in Fig. 8. (This configuration differs from the pre-configured test case, which has uniform drag at 3 bars, and from Schneider and Liu, 2009, who only had drag polewards

of 16°.) This model is configured entirely with namelist parameters or Python dictionaries from the Isca master model, without need for recompiling. Extensions and variations of this type of model may be (and have been) configured — the addition of moisture (with a moist convection scheme appropriate for a hydrogen atmosphere), setting the lower boundary to be at a much higher pressure, different drag formulations, and so forth, and our own investigations continue.

## 8.2 Slowly-rotating terrestrial planets

To illustrate some of the capabilities of Isca as an idealized model of terrestrial planets other than Earth, we show the results of simulations performed with a thermal-damping forcing, first reducing the planetary rotation rate $\Omega$ (relative to Earth, $\Omega = \Omega_E$) by a factor 20, then increasing the atmospheric depth (surface pressure $p_s$). This corresponds to moving the model in the direction of Titan and Venus: Titan's rotation rate is about 1/16 that of Earth, its diameter is about 0.4 of Earth and its surface pressure is 1.5 times larger; Venus has a similar radius to Earth but its rotation rate is 243 times less and its surface pressure (92 bars) is almost two orders of magnitude larger. Although the model we use here is highly idealized the results do exhibit some key features of the these atmospheres.

Figure 9 shows the time- and longitudinally-averaged zonal wind for a model Earth (panel (a)) and for planets rotating at 1/20 the rate of Earth with surface pressures $p_s$ = 1, 7.9 and 92 bars. (The first case is essentially a Held–Suarez version of Earth and the second case is similar to one in Pinto and Mitchell (2014).) In the three cases with reduced rotation the circulation between the zonal jets is a Hadley cell that nearly conserves momentum in its upper branch and extends further poleward than on Earth, as expected.

The temperature forcing has the same equilibrium state $T_{eq}(\theta, p)$ (with no diurnal or seasonal variation) in all four cases, and produces a tropopause at about $p = 200$ hPa. In case (b), there is a weakly superrotating layer at this level. For the progressively deeper simulations (panels (c) and (d)) the same number of pressure scale heights was used (in order to limit wave-breaking; other than grid-scale $\nabla^8$ hyperviscosity, the only momentum damping deployed here is the near-surface Rayleigh damping) but the top of the simulated atmosphere was still above the tropopause level. In the deeper cases, the superrotating layer is strengthened to zonal wind speeds similar at the equator to those at the core of the high-latitude jets, and these are fastest in the deepest case. Similar experiments with a zonally-symmetric model (not shown) do not exhibit equatorial superrotation, as expected

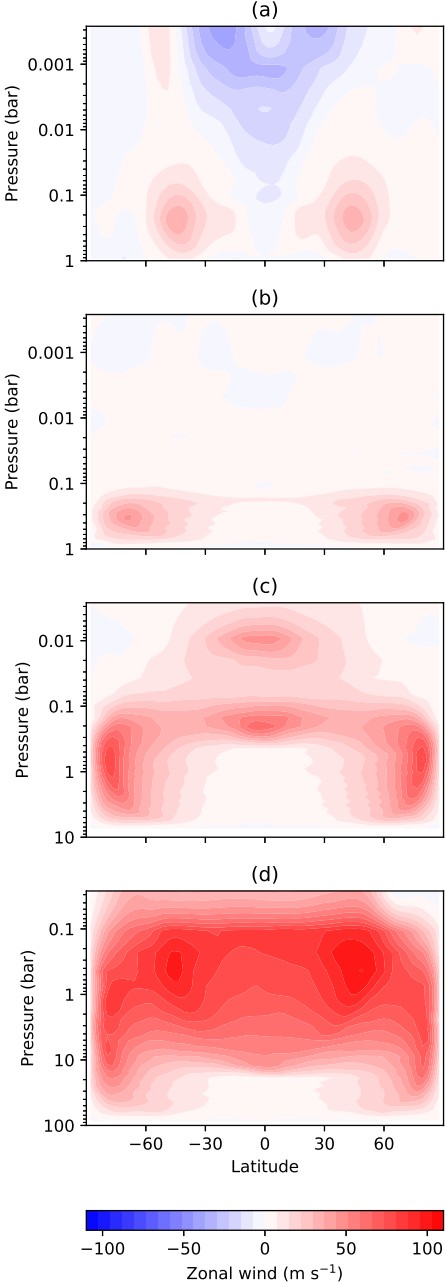

**Figure 9.** The time- and longitudinally-averaged zonal wind, in m s$^{-1}$, versus latitude and pressure level, for (a) $\Omega = \Omega_E = 7.3 \times 10^{-5}$ rad s$^{-1}$ and $p_s = 1$ bar, (b) $\Omega = \Omega_E/20$ and $p_s = 1$ bar, (c) $\Omega = \Omega_E/20$ and $p_s = 7.9$ bar, (d) $\Omega = \Omega_E/20$ and $p_s = 92$ bar. These results are obtained with 30 unequally spaced sigma levels and T42 horizontal resolution.

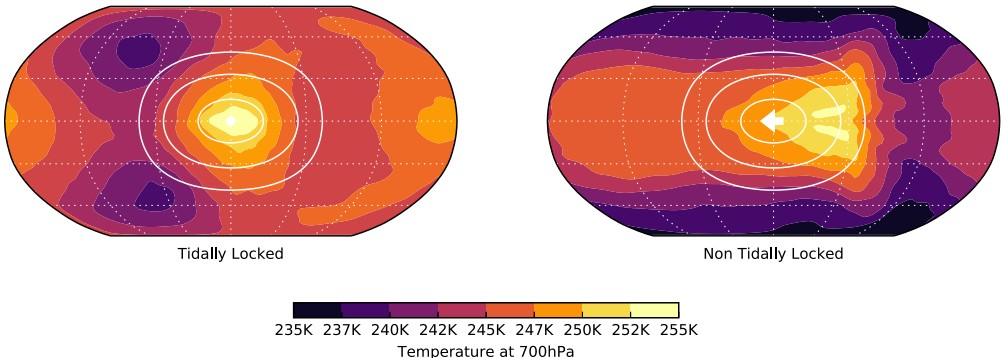

235K 237K 240K 242K 245K 247K 250K 252K 255K
Temperature at 700hPa

**Figure 10.** Experiments comparing the atmospheric dynamics on tidally-locked and non-tidally-locked exoplanets, using a primitive equation model with forcing via thermal relaxation to a specified field. Filled colour contours show the temperature at 700 hPa and white contours show the location of the forcing. For the non-tidally-locked case the substellar point is shown with a small white arrow denoting is direction of passage, which is to the left, here with a velocity of $25\,\mathrm{ms}^{-1}$.

since eddy motion is required to create an angular momentum maximum (Hide, 1969; Vallis, 2017).

There is observational evidence from both Titan and Venus to suggest a wide Hadley cell and strong superrotation aloft. For example Sánchez-Lavega et al. (2008) found in Venus
Express data that the zonal winds on Venus at the cloud level were approximately 60–100 m $\mathrm{s}^{-1}$ (the higher figure roughly at the tropopause level) from the equator out to about 50–60°, and then decreased to the pole as is also seen here. They also found the peak meridional winds to be at 55°S; this latitude is well poleward of the Hadley cell on Earth. However, it has proven notoriously difficult to quantitatively reproduce Venusian winds, even with
comprehensive Venus models, and our investigation of the parameters that determine these winds, and with more nearly Venusian parameters, will be reported elsewhere.

### 8.3   Exoplanets

Within Isca it is straightforward to change orbital parameters to map out some of the possible circulation regimes that could exist on planets outside our Solar System, using either the
simplified or full radiative transfer schemes, or thermal relaxation. Here we show an example using the latter to model the changes in circulation as a planet passes from being tidally-locked – that is, the same face is always pointed to its host star – to having a diurnal cycle, which may be of varying length. The length of the diurnal cycle, $T_{\mathrm{sol}}$, is given by the relationship

between rotation and orbital rate

$$T_{\text{sol}} = \frac{2\pi}{\Gamma - \Omega},$$

(15)

where $\Gamma = 2\pi/P_{\text{orb}}$ is the orbit rate and $\Omega$ the rotation rate of the planet. The longitude of the substellar point – equivalent to the longitude of midday on Earth, $\lambda_*$, is then

$$\lambda_*(t) = 2\pi \frac{t}{T_{\text{sol}}} = (\Gamma - \Omega)t.$$

(16)

For a tidally-locked planet, orbital and rotation rate are equal and the substellar point remains fixed in time.

We have configured the thermal relaxation parameters (of the three-dimensional primitive-equation dynamical core) to a longitudinally asymmetric heating profile that moves according
to (16), and the planetary rotation rate and the planetary orbital rate (around its sun) are then chosen to give tidally and non-tidally locked configurations. These configurations can be made with the Python front end. Example results are shown in Fig. 10 for a planet that is Earth-like in size, atmospheric density and composition. The model is run to a statistically-steady state in each case with a rotation rate, $\Omega = 1 \times 10^{-5}\,\text{s}^{-1}$, that is approximately 10 times slower than
Earth. The equator to pole temperature gradient of $\Delta T = 60\,\text{K}$ means that the external thermal Rossby number of the system is large, $\text{Ro}_\text{T} = (R\Delta T)/(2\Omega a)^2 \simeq 100$ (where $R$ is the ideal gas constant). The tidally-locked configuration shows a pattern resembling a Matsuno-Gill solution (also seen in Merlis and Schneider, 2010 and Showman and Polvani (2011)), with Rossby lobes westward and poleward of the heating, and with a maximum temperature (the
hotspot) at the sub-stellar point. Interestingly, in the non-tidally locked case the hotspot is not co-located with the sub-stellar point and may lead or lag, as was discussed using shallow water dynamics by Penn and Vallis (2017).

Isca is not limited to using a thermal relaxation scheme for such exoplanets; the array of parameterizations available allows for increasing levels of complexity depending on the data
available and the user's preference. Isca could be configured to study a specific star-planet system using a gray or multi-band radiation scheme, parameterized for the observed stellar output and atmospheric composition of the star and planet, respectively, and with topography, a continental land mass and an ocean.

## 9   Python Interfaces

In addition to the many model options provided in Isca, we have endeavoured to make the model framework as easy as possible to use and configure. To that end we have interfaced

**Land & Hydrology**

1. Aquaplanet (no land).
2. Idealized or realistic continental outline and topography.
3. Bucket hydrology.
4. Evaporative resistance.

**Ocean**

1. Slab mixed layer.
2. Q-fluxes: idealized or targeting an arbitrary SST distribution.
3. Simple sea (and land) ice.

**Convection**

1. Convective relaxation: simplified Betts-Miller, or Betts-Miller.
2. Mass flux using relaxed Arakawa-Schubert.
3. Large-scale only.

**Software**

1. Python front end: for running the model, setting model configuration and parameters.
2. Fortran and message-passing internals, GFDL FMS infrastructure.
3. Git-based, open source repository.

**Primitive Equation Spectral Core**

1. Three-dimensional.
2. Zonally-symmetric.

**Infra-Red Radiation**

1. Gray fixed optical depth.
2. Gray with $H_2O$ and $CO_2$.
3. Two-band for IR with $H_2O$ and $CO_2$.
4. RRTM comprehensive scheme.

**Planetary**

1. Arbitrary atmospheric mass, rotation rate, gravity.
2. Solar input dependent on obliquity, eccentricity, solar constant.
3. Configurable diurnal and seasonal cycles, tidally locked, spin resonant, etc.

**Thermal Relaxation**

1. Held–Suarez.
2. Astronomical and radiatively determined, radiative-convective equilibrium temperature.

**Solar Radiation**

1. Transparent atmosphere.
2. Specified absorption.
3. RRTM: comprehensive, composition dependent.

**Figure 11.** A summary of some of the main options currently available in Isca.

the model's underlying Fortran code with Python. The Python front end that is included provides a way to define, build and run experiments that are easy to reproduce and rerun. More details are accessible in the online documentation, but here is a brief summary of the notable features.

5    1. A full experiment can be configured from a single Python script. Namelist parameters and diagnostic output configuration are provided using native Python dictionaries and objects, so that the entire experimental set-up can be specified from a single document.

2. The Python scripts provide support for parameter sweeps; that is, the user may perform several experiments by varying one or more parameters from a single run script.

3. The scripts simplify building and running on different architectures, as the experiment scripts are independent of the specific build requirements of the computational architecture. Once the model is configured to build on a computer, all Python-based experiments can be run on that machine.

4. The scripts are version-control aware: experiments can be run using a specific commit or version of the codebase, so that if the experiment needs to be re-run in the future to reproduce some results, the exact same code will be used.

5. Using these scripts, Isca has been run on multi-core Linux workstations and on the University of Exeter supercomputer, and on clusters and supercomputers elsewhere. Porting to other traditional architectures should be fairly straightforward, given the availability of an appropriate Fortran compiler, a Message Passing Interface and Python.

The scripts are currently agnostic to Python 2.7 and 3.5, although in future Python 2.7 may be deprecated if needed to maintain operability.

## 9.1 Post processing and diagnostics

We provide various post-processing capabilities, mainly in Python, although the user would of course be free to design their own. Diagnostics available within Isca itself include Python software to interpolate model output to a higher resolution and then restart the model at higher resolution, and an interpolator to produce output on pressure levels.

Current users of Isca have constructed eddy fluxes of heat and momentum, a ray-tracing package to construct group velocities and plot ray trajectories for Rossby waves and, of course, the software required to read the NetCDF output from the models and construct the plots in this paper, often making use of the xarray toolkit (Hoyer and Hamman, 2017). The post-processing software is not packaged within Isca itself but some packages may be available on individual user repositories, and a community repository may be set up in future.

## 9.2 Test cases

Although the framework is not intended to be used as a black box, we do provide a number of test cases that will run 'out of the box' using the Python front end and with minimal configuration by the user. These include: (i) The Held–Suarez test case; (ii) A dry model

case using astronomically and radiatively determined thermal relaxation temperature fields, with seasons; (iii) A moist aquaplanet with gray radiation, with or without seasons; (iv) A moist aquaplanet with RRTM radiation and specified ozone, as in the MiMA model; (v) A case with a simple continent using bucket hydrology and RRTM radiation; (vi) Cases with variable $CO_2$ concentrations using either the gray and RRTM radiation schemes; (vii) A giant planet, similar to Jupiter. (viii) Cases with realistic continents with either Q-fluxes or prescribed SSTs. Axi-symmetric versions of some of these cases are, where sensible, also available.

We also provide a trip test, whereby following some new software implementation (e.g., a new commit on the git repository) a suite of model tests, corresponding to many of the cases above, can automatically be performed to make sure that the new software has not introduced any unwanted behaviour, and that runs are bitwise identical with previous model versions where appropriate.

## 10  Concluding Remarks

In this paper we have presented a framework for the construction and use of global circulation models of varying levels of complexity, from dry dynamical cores to more realistic moist models with full radiation schemes as well as land, mixed layer oceans and topography. We have also presented a few examples of models within that framework, and we hope that other users may be motivated to use the framework to construct more such models. The models that one is currently able to straightforwardly configure connect to, but fall a step shy of, the truly comprehensive models used for quantitative climate projections. Construction of models of other planetary atmospheres, with different compositions other parameters, may be straightforward or not depending on the planet and the level of complexity desired. A summary of the main features and options in our framework is provided in Fig. 11.

Compared to a truly comprehensive climate model (of which there are many), significant missing features are a sophisticated land-surface model, interactive clouds and a dynamical ocean. An idealized ocean-atmosphere coupled model, in a similar framework, was previously presented by Farneti and Vallis (2009) and we hope to incorporate a similar capability into Isca, as well as an idealized capability for interactive cloud modelling, in future. Note, though, that our goal is not to provide another comprehensive model, nor to prescribe a single hierarchy; rather, it is to provide a means whereby a complex system may be easily modelled in different ways, with different levels of complexity, so providing a nearly continuous

pathway from comprehensive numerical modelling to conceptual modelling and theory for Earth and planetary atmospheres.

An ambitious goal in the climate sciences and, increasingly, in the planetary sciences, is to construct a so-called traceable hierarchy, in which each model is connected to another of greater or lesser complexity, enabling one to pass from a state-of-the-art comprehensive model to a very simple model in a sequence of (non-unique) connected steps. Although we have not fully enabled that program we have made some steps toward it, in the restricted context of the global circulation of planetary atmospheres.

*Code availability.* A general introduction to the framework can be found at http://www.exeter.ac.uk/isca. The code (v1.0 and later versions) is publicly available from github at https://github.com/ExeClim/Isca, and v1.0 is also available in the supplementary information to this article. Use of the github site is recommended for most users.

*Author contributions.* All authors have contributed to the general development of the software and to the writing of this paper. Among other contributions, Stephen Thomson implemented Q-fluxes, the Jovian configuration, the simple land and ice models, code allowing mid-stream resolution changes, the trip tests, and ported RRTM to Isca; Ruth Geen implemented the two-plus-one-band radiation scheme and bucket hydrology, and contributed to the continental set up; James Penn designed and implemented the Python configuration tools and front-end (which many other components use), and constructed many of the planetary-atmospheric and exoplanet options; Penelope Maher contributed an initial model set up and website, and ported the RAS scheme to Isca; Greg Colyer implemented a zonally-symmetric dynamical core, a Venusian configuration, and has managed the git repository; Alex Paterson implemented the astronomically and radiatively based dry thermal relaxation scheme; Marianne Pietschnig tested Isca with very idealized continents and bucket hydrology; Martin Jucker and Edwin Gerber developed the MiMA model with RRTM, from which Isca has drawn; and Geoffrey Vallis envisioned and has overseen the project as a whole.

*Competing interests.* The authors declare no competing interests.

*Acknowledgements.* This work was funded by the Leverhulme Trust, NERC (grant NE/M006123/1), the Royal Society (Wolfson Foundation), EPSRC, the Newton Fund (CSSP project) and the Marie Curie Foundation. We thank Qun Liu, Dann Mitchell and two anonymous reviewers for their comments. We also acknowledge the model foundation and software infrastructure from GFDL, and numerous colleagues around the world for making their software publicly available.

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
