# Peer review of "Isca, v1.0: A Framework for the Global Modelling of the Atmospheres of Earth and Other Planets at Varying Levels of Complexity"

_Geoscientific Model Development, 2017_

## Short Comment (SC1) · 13 Dec 2017

This is a very valuable contribution to the literature, and in my view, certainly fits well in this journal. The authors should be commended on the effort that has gone into this, as well as the openness of the code. I have a few comments, mainly as points of clarification that do not change any of the messages of the paper.

General comments:

1. It might be a lot of effort, but I think this paper would benefit from having an appendix of terms at the end. The issues in this paper deal heavily with radiative processes, and

dynamical process. Normally, the reader will be an expert in only one of these areas, and so may get lost in the other. For my part, terms like 'correlated-k radiation scheme' are a little abstract. Alternatively, a brief short explanation for these terms in the main text may help.

2. Python can be a nightmare for version control, with syntax changing rather dramatically between versions. Will the Python wrappers be frozen at a certain version, or continuously updated? I think this is different to the point 4 on page 15, which deals with a slightly different issue, but I might be mistaken.

Minor clarifications/changes:

1. The affiliation order is wrong for Ed and Martin (2 comes before 3!).

2. The name 'Isca' is a little confusing, it is not an acronym, but neither is it a real word (as far as I know). A couple or words explaining why it is called this would help.

3. P2 (left column). Is not another very valuable part of this framework just to explore interesting GFD problems, some of which we really are struggling to answer (for instance the strat-trop couple mechanism). So studying the GFD of other planets can help with Earth.

4. P5 (line 44): 'The shortwave band treats all solar radiation', I don't really follow this sentence, how is all radiation treated when it only covers the shortwave?

5. P6 (lines 38-39): What is meant be a 'reasonable' stratospheric and polar vortex? I'd argue that some CMIP5 models don't have a reasonable representation of these.

6. Section 4.4: Is it worth highlighting how important IR can be for planetary atmospheres. For instance, the O3 in Earth's stratosphere, or organic hazes in Titan's atmosphere. Without the IR, you get a completely different response. It might be worth making this point explicitly.

7. Section 6.2: This might be a naïve question, but is water ice treated differently from

other ice's such as CO2 ice? Is there a reason to believe they might be different, in for example, their surface roughness?

8. P12 (lines 5-6): 'Here we make more modest changes...' this sentence is not clear to me. Is the 'more' attached to 'modest' or 'changes', i.e. are you making modest changes to the rotation rate and surface pressure?

9. P13 (lines 5-6): 'We have configured the thermal...' is this something that can be done through the Python interface? It sounds like it is a more fundamental code change.
* * *

---

## Author Comment (AC1) · 15 Dec 2017

Many thanks for your very helpful comments. There is really nothing that we disagree with, and we'll provide a longer response when we revise the paper. We will ponder the Python/revision control issue that you raise.

Also, FYI, Isca is the Roman name for Exeter, and it also means running water, so it seemed an appropriate name for the framework. See also https://execlim.github.io/IscaWebsite/

2017.

---

## Referee Comment (RC1) · Anonymous Referee #1 · 17 Dec 2017

As its title suggests, this manuscript describes a modeling framework, based around a well established dynamical core, for simulating the 3D time-dependent circulation of Earth-like atmospheres over a wide range of conditions and with varying degrees of sophistication. The motivation is well described as seeking a traceable hierarchy of modeling tools that range from highly idealized, hypothesis-testing models all the way up to quantitatively realistic simulation models that can be compared directly with observations - at least for the Earth itself.

Although not unique, this looks to be a potentially useful set of modeling tools that has been designed to be relatively easy to install and use on a wide variety of platforms,

and yet is based on some well established and well regarded models derived from the Princeton GFDL FMS model suite. An attractive feature is the availability of a set of Python programs, presumably designed and written by the Exeter group of post-docs and students, that can be used to configure the model, set its parameters and run it. This should make it accessible to relatively inexperienced users.

The manuscript itself describes the motivation and model formulation reasonably thoroughly and clearly, and includes a useful set of examples to illustrate what the model framework is capable of. My detailed comments below are mainly to seek clarification and to note some minor cosmetic errors and clumsy phrasing. But once these are corrected and addressed, the paper should be ready to be published.

The main substantive comment is that, although almost all the relevant parameterizations are described and discussed, I was surprised to see nothing explicitly mentioned about implementing topography, other than to say that it is included. This would normally be considered as one of the properties of the land surface that needs to be specified alongside the other surface boundary conditions, and perhaps should be included in an extra subsection in Section 5?

I was also somewhat less than convinced at the simulations of both Jupiter-like and Venus-like planets, neither of which looks more than superficially like their Solar System counterparts (the "Jupiter-esque" jets seem too weak and without an equatorial jet, for example, unlike the Schneider & Liu model comparators mentioned in the text, let alone observations). The Venus-like case also only seems to exhibit tropical super-rotation at very high altitude, near the model lid, which differs markedly from Venus itself. But perhaps this is expecting too much of a very idealized model! However, at a rotation speed 1/20 that of the Earth, wouldn't the comparison be expected to look closer to Titan than Venus?

P.1 line 4 There is more than one "Geophysical Fluid Dynamics Laboratory" in the world(!), so this should specify that you are referring to the one in Princeton, USA.

P.5 line 4 Not very satisfactory to refer to a paper still "in preparation" (Paterson & Vallis 2017).

P.5 line 16 Eq (2) This presumably only applies in the troposphere, as the optically thin stratosphere would become isothermal in radiative equilibrium. This expression is presumably a linearized approximation, which should be made clearer.

P.5 line 21 Should this be referring to equation (2) not (1)? Also delete "the" on line 22.

P.7 line 7 "the a" should be one or the other?

P.9 line 5 Is it really the case that the full RRTM cannot cope with more extreme conditions? Can you not recalibrate or extend the k-coefficients to adjust for more extreme situations?

P.10 Section 5 - An absence of discussion of topography here seems surprising. Perhaps add another subsection?

P.16 line 19 & ff Wording here is rather clumsy. Something like "where $\sigma_b$ represents the lowest level at which the drag is applied, $\sigma_t$ is the uppermost level and….." would be better.

P.17 and Figure 9 and associated text. The vorticity is perhaps not the easiest field to interpret quantitatively for the "Jupiter-esque" case. Perhaps also show a latitude-height section of zonal velocity?

P.17 and Figure 9 - Why are the jets closed at ∼0.1 bars in the Venusian case? Are they also closed for p_s = 7.9 and 62 bar? Showing u with a vertical scale that is linear in pressure is not very clear….perhaps linear in log p or height would be better? How many vertical levels in each case?

P.18 line 2 Parentheses around references needs attention.

P.20 section 9.1 Are there any plans to make post-processing routines more widely available? Perhaps via a user forum?

P.22 line 2-3 A word seems to be missing here to make sense.

P.22 line 10 "it is TO provide a means" (word missing?).

---

## Referee Comment (RC2) · Anonymous Referee #2 · 18 Dec 2017

Various concerns have already been raised by the other two referees, so I will not repeat them here. However, I will ask that the authors properly give credit to past studies that are relevant. As it stands, the manuscript reads like the idea of a flexible climate modelling suite to study Solar System bodies and exoplanets is a completely novel idea, when in fact workers in the various communities have toiled for years.

Merlis & Schneider (2010, Journal of Advances in Modeling Earth Systems): these authors perform a rather comprehensive suite of simulations of Earth-like aquaplanets, exploring the differences between an Earth-like rotation rate and a tidally-locked configuration.

[Figure]

Heng, Menou & Phillipps (2011, MNRAS, 413, 2380): these authors use the FMS and build on the Held & Suarez (1994) model, extending it to a tidally-locked, Earth-like configuration as well as hot-Jupiter-like configurations.

Showman & Kaspi (2013, Astrophysical Journal, 776, 85): these authors explore GCMs for brown dwarfs and directly imaged exoplanets, which are close cousins of our Jupiter.

Kaspi & Showman (2015, Astrophysical Journal, 804, 60): these authors explore a suite of GCMs for what they call "terrestrial exoplanets", e.g., varying rotation rate, insolation.

Mendonca et al. (2016, Astrophysical Journal, 829, 115): these authors implement the Satoh+ HEVI scheme and an icosahedral grid on GPUs to construct a flexible GCM from scratch, not unlike what the present study wishes to do.

---

## Author Comment (AC2) · 20 Dec 2017

Many thanks to both reviewers for their comments. There is nothing that we really disagree with, and we will take them all into account, clarify statements, add references etc., if and when we are invited to submit a revision.

Best wishes for the Holiday Season.

---

## Author Response (AR1)

**Response to Reviewers' Comments, GMD D2017-243**

We thank all three reviewers for their comments. We have tried address all of them, and our response is below. Reviewers' comments are italicized and response is in roman. We have also made a number of minor textual changes throughout and Figure 11 has been redrafted, but its content is the same.

**Response to Interactive Comment from Dann Mitchell**

Many thanks for your remarks. Our response is as follows.

**General Comments**

1. *The paper would benefit from having an appendix of terms...or alternatively a brief short explanation of terms in the main text.*

   We've gone with the option of explanations in the text. The correlated-k term was specifically mentioned, and here we have added: 'The 'correlated-$k$' method, with $k$ being the absorption coefficient, is a means to efficiently calculate radiative transfer over a broad spectral range by collecting together wavenumber intervals with similar spectral properties and by supposing that these spectral properties are correlated from one level to another. A relatively small set of absorption coefficients can then be chosen to be representative of the absorption coefficients for all frequencies, leading to an enormous speed-up over line-by-line calculations and much better accuracy than traditional band methods that more simplistically just group together similar wavenumbers.' The references given in the text provide more detail.

2. *Python can be a nightmare for version control with syntax changing dramatically...Will the Python wrappers be frozen at a certain version? [Is] this different from point 4 on page 15?*

   That is an interesting point. Python did indeed change from Python 2 to Python 3 causing some disruption. There is an informative blog about this by one of the Python core developers at:

   https://www.curiousefficiency.org/posts/2014/08/python-4000.html

   Essentially, Python 4 will not change significantly from Python 3, and in any case Python 4 will not appear for quite some time — it now appears that the release after Python 3.9 will be Python 3.10. Our scripts currently support Python 2 and Python 3, but it may be that Python 2 support is dropped at some stage if there are advantages in having only Python 3, and there will be an official end-of-life of Python 2 in 2020. After that we don't expect there to be significant changes in Python syntax for the foreseeable future, but we would try to keep up to date. On that timescale changes in Fortran may also affect compilation of the core code. This issue is different from

point 4; there we are remarking that the Python scripts have the ability to extract an old version of the model from the repo and compile and run it.

**Minor Comments**

1. *Affiliation* – fixed.

2. *Explanation of Isca* – added in a footnote on P. 3.

3. *Studying GFD of other planets can help with Earth* – Of course we agree, and think it is clear.

4. *Shortwave band treats all solar radiation.* – We are not sure what the problem is with this statement. We don't say it treats all radiation, just all *solar* radiation, and shortwave is a common term for solar radiation.

5. *A reasonable stratospheric polar vortex.* – We have removed 'reasonably realistic'.

6. *Importance of IR for planetary atmospheres.* – Yes, right. nearly all models, including idealized ones, have IR included, and here we are making the explicit point about the two-band scheme being an improvement over a grey model.

7. *Water ice vs CO$_2$ ice.* – Here we mean water ice and this is clarified. We haven't tried to model CO$_2$ ice, such as might be found on Mars, with the model.

8. *More modest changes.* – we rephrase to make this clearer.

9. *We have configured the thermal relaxation parameters.* – These are fundamental changes in the Fortran code. However, now that they are done, the user may change the parameters using the Python interface. Now clarified.

**Response to Interactive Comment from Anonymous Referee #1**

Many thanks for your remarks. Our response is as follows.

**General Comments**

*Nothing is explicitly mentioned about topography.*

Thanks for pointing that out. We have added a subsection to Section 5, (section 5.3) as the reviewer suggested.

*I was less convinced at the simulations of Jupiter-like and Venus-like planets.*

Fair enough, and there are two remarks we wish to make here.

*(i)* Most importantly, in this paper we are just trying to demonstrate the capabilities of the framework, rather than discussing the science issues as to why, for example, the jets are where they are or presenting simulations that compare well with reality. That is an ongoing (and difficult!) task and we hope to present such work in subsequent papers in the disciplinary literature.

*(ii)* Second, and more specifically, we have clarified our work in this paper (Section 8), and don't claim we are simulating Venus or Jupiter – we now talk about 'slowly rotating planets' and 'giant planets' for example. Indeed Titan is a better example for our slow rotation rate, although Venus is a better example for the increase in mass. Regarding the Venus-like case that only seemed to produce tropical super-rotation near the model lid: this may have been an unintended consequence of our graphics. We now show them using log-pressure co-ordinates (figure 9), as per a later comment by this reviewer. Regarding Jupiter, note that the Figure 8 showed vorticity, not velocity, and the jets are not especially weak. Other simulations we have done do give equatorial jets of a different nature and we will be presenting this work in future.

**Specific Comments**

- *There is more than one Geophysical Fluid Dynamics Laboratory.*
  Quite so! The one founded by Raymond Hide is particularly revered by some of us, and there may be others. We've specified we mean the Princeton one.

- *P5. Reference to a paper in preparation.* – Removed.

- *P5 Eq (2) applies only in troposphere, and is presumably linearized.* Yes, it only applies in the troposphere. It can be extended upward isothermally, and we have also clarified this. The Newtonian relaxation scheme itself is a linearization and we explicitly note that on P7.

- *P5 line 21 and line 22.* – Fixed.

- *P7, line 7. 'The a'* – Fixed.

- *P9, Can RRTM not cope with more extreme conditions by extending the k-coefficients?*
  Yes, it can in principle, as can any correlated-k scheme. However, we have not done this, and are not aware that others have either. We've clarified this at the end of section 4.3. Also, we have added a brief description of what a correlated-k scheme at the beginning of this section, in response to another reviewer's comments.

- *P10, section 5. Needs a section on topography* Agreed, and we have added a subsection 5.3.

- *P 16, line 19., wording is rather clumsy.* Thanks for the close reading. We have rephrased along the lines you suggested.

- *P 17, Fig 9 and text. Vorticity not the easiest field to interpret.* We see your point, but please note our response to a related remark of yours in the general comments above.

- *P 17, figure 9 – jets in the Venusian case* We no longer call it the Venusian case and we now show a log pressure plot (figure 9) which clarifies these questions, and we state how many levels there are. Again note our response to your remark in the general comments above.

- *P18, line 2 parantheses wrong.* – Fixed.

- *P20, plans for a user forum?* That is a good idea and something that we have discussed, although at the moment we have no specific plans. If and when we have more external users we will certainly revisit that issue.

- *P22, a word missing to make sense.* – We slightly rephrased this sentence.

- *P22, word 'to' missing.* – Fixed. Many thanks for noticing these things.

**Response to Interactive Comment from Anonymous Referee #2**

Many thanks for your remarks. Your main comment was that we had not properly given credit to past studies and you gave some examples. Of course that was not our intent. We might note that in the very first paragraph of the original paper we did write:

> Various models at different levels of complexity have indeed been constructed (Frierson et al., 2006; Mitchell et al., 2006; O'Gorman and Schneider, 2008; Blackburn and Hoskins, 2013; Joshi et al., 2015, to name but a few).

To this list we have now added Fraedrich et al (2005, the useful PUMA model). We have added a number of other references to papers that have a more planetary aspect, so that the introduction now contains:

> A variety of models at different levels of complexity have in fact been constructed. Thus, to name but a few, Fraedrich et al. (2005b); Frierson et al. (2006); O'Gorman and Schneider (2008); Blackburn and Hoskins (2013) and Joshi et al. (2015) provide models of Earth's atmosphere that are simplified in some way compared to a full GCM (of which there are a very great many). Similarly, regarding planetary atmospheres and again giving a limited sample, the Planet Simulator is a sibling of the PUMA model for planetary atmospheres (Fraedrich et al., 2005a); the SPARC model (Showman et al., 2009) uses the dynamical core of the MIT GCM but adds a more general radiation scheme appropriate for planetary atmospheres; the GFDL system has itself been used in a number of Earth and planetary settings (e.g., Mitchell et al., 2011; Schneider and Liu, 2009, and others); the UK Met Office Unified Model has been configured in various ways for both terrestrial exoplanets and hot Jupiters (Mayne et al., 2014; Boutle et al., 2017); the THOR model (Mendonca et al., 2016) solves the deep non-hydrostatic equations (as does the Unified Model) on an icosohedral grid and is designed to explore a range of planetary atmospheres; and CliMT (https://github.com/CliMT/climt) aims to provide a flexible Python based climate modelling toolkit. A number of quite comprehensive models, targeted at specific planets and similar in some ways to full GCMs of Earth, have also been developed.

We have mainly focussed on flexible models (rather than on papers that take a particular model and use it over a range of parameters) and we haven't tried to reference comprehensive models or models of particular planets. Merlis and Schneider is referenced later in the paper.

[revised manuscript text omitted]